# Recovery of Platinum Group Metals from Spent Automotive Catalysts Using Lithium Salts and Hydrochloric Acid

**DOI:** 10.3390/ma14226843

**Published:** 2021-11-12

**Authors:** Shunsuke Kuzuhara, Mina Ota, Ryo Kasuya

**Affiliations:** 1National Institute of Technology, Sendai College, 48 Nodayama, Medeshima-Shiote, Natori 981-1239, Miyagi, Japan; kuzuhara@sendai-nct.ac.jp (S.K.); ota.mina.p2@dc.tohoku.ac.jp (M.O.); 2Global Zero Emission Research Center, National Institute of Advanced Industrial Science and Technology (AIST), 16-1 Onogawa, Tsukuba 305-8569, Ibaraki, Japan

**Keywords:** leaching, PGMs, LIBs, recycling, fluorine immobilization

## Abstract

The recovery of platinum group metals (PGMs) from waste materials involves dissolving the waste in an aqueous solution. However, since PGMs are precious metals, their dissolution requires strong oxidizing agents such as chlorine gas and aqua regia. In this study, we aimed to recover PGMs via the calcination of spent automotive catalysts (autocatalysts) with Li salts based on the concept of “spent autocatalyst + waste lithium-ion batteries” and leaching with only HCl. The results suggest that, when Li_2_CO_3_ was used, the Pt content was fully leached, while 94.9% and 97.5% of Rh and Pd, respectively, were leached using HCl addition. Even when LiF, which is a decomposition product of the electrolytic solution (LiPF_6_), was used as the Li salt model, the PGM leaching rate did not significantly change. In addition, we studied the immobilization of fluorine on cordierite (2MgO·2Al_2_O_3_·5SiO_2_), which is a matrix component of autocatalysts. Through the calcination of LiF in the presence of cordierite, we found that cordierite thermally decomposed, and fluorine was immobilized as MgF_2_.

## 1. Introduction

Platinum group metals (PGMs), which include Pt, Pd, Rh, Ir, Ru, and Os, are used in various fields, including catalysis. PGMs are commonly employed in catalysts for the treatment of automobile exhaust gases. In 2020, the gross demand recorded for PGMs was 228.0 tons of Pt, 311.3 tons of Pd, and 31.7 tons for Rh [1]. However, only 153.8, 191.6, and 18.9 tons of Pt, Pd, and Rh, respectively, were produced from mines; thus, the demand was not completely met. According to reports [1,2], the demand for PGMs has been met through mining and recycling for more than a decade.

Since PGMs are rare and expensive elements, their recovery and recycling from waste materials, including automotive catalysts (autocatalysts), have received growing attention. The International Energy Association (IEA) provided a forecast for electric vehicles (EVs) and assessed the impact that the coronavirus disease (COVID-19) will have on this market [3]. In this context, the term EV refers to both battery electric vehicles (BEVs) and plugin hybrid electric vehicles (PHEVs). According to the Stated Policies Scenario (STEPS), which presumes that policies announced thus far will be realized, the number of EVs worldwide will increase from 8.1 million recorded in 2019 to 140 million by 2030 (an increase of ~30% annually). A total of 7.2 million EVs, which were passenger light-duty vehicles (PLDVs), existed in 2019, and this figure is expected to reach 43 and 119 million by 2025 and 2030, respectively.

Even in the “bright” future predicted by the STEPS, EVs will only account for approximately 10% of the PLDVs. Automobiles that are not BEVs or fuel-cell vehicles are equipped with internal combustion engines (ICEs), and it is expected that ICE-equipped vehicles will still account for approximately 91% and 84% of the total automobile sales in 2030 and 2040, respectively [4]. Since LDVs have a life expectancy of ~15 years [5], it is expected that PGM recovery from used autocatalysts will continue in the future.

The recovery of PGMs from waste materials usually involves dissolving the waste in an aqueous solution, such as an acid solution. Subsequent mutual separation processes, such as solvent extraction, precipitation separation, or ion exchange, target the PGM ions present in solution. However, since PGMs are precious metals, they are sparingly soluble in protonic acids (e.g., hydrochloric acid (HCl)). Therefore, on an industrial scale, acids containing oxidizing agents such as chlorine gas and aqua regia (a mixture of HCl and nitric acid) are employed [6,7]. While these oxidizing agents have great potential to oxidize PGMs, they are highly toxic and corrosive.

Previously, to avoid the use and generation of toxic gases, we proposed a novel dissolution process for PGMs based on composite oxides [8,9,10,11,12,13]. This novel process enables the PGMs to be oxidized (ionized) through their calcination with alkali metal salts, thereby avoiding the requirement of highly toxic oxidizing agents in the dissolution process. Li salts are especially suitable for recovering PGMs from waste materials as they universally form composite oxides. It is also well known that sodium salts such as NaHSO_4_ and Na_2_O_2_ are used in alkali fusion for the formation of complex oxides. In contrast, our new process utilizes oxygen from air as an oxidizing agent. In addition, solvent extraction using HCl can be applied in the existing process.

Currently, lithium-ion batteries (LIBs) account for up to 71% of the demand for Li [14]; LIBs are widely used in EVs and in small electronic devices, such as smartphones and laptop computers. Furthermore, the widespread use of LIBs has resulted in increased demands for Li. In the STEPS scenario from the IEA, it is expected that the demand for Li in 2040 may be 13 times higher than that of 2020 [15].

Since the several hundred charge/discharge cycles of LIBs result in their deterioration over a number of years [16,17], it is expected that they will be discarded at approximately the same rate as that of their production [18]. Currently, the recycling rate of used LIBs is less than 1% [15,19], with 95% of them reaching landfills [20]. This improper disposal of used LIBs leads to environmental deterioration and renders the proper disposal and recovery of LIBs a necessity [21,22,23]. In order to recover metals from LIBs, we applied a carbon reduction method, which utilizes carbon powder as a reducing agent. Through this dry process, we succeeded in separating and recovering Li and Co from an LIB cathode model (LiCoO_2_) [24]. When targeting waste LIBs, heating is an appropriate, inexpensive treatment; however, in such situations, it is necessary to control the levels of F originating from the electrolytes (e.g., lithium hexafluorophosphate, LiPF_6_) and the binders (e.g., polyvinylidene difluoride, PVDF) of these batteries. For example, LiPF_6_ decomposes due to the moisture in air to produce lithium fluoride (LiF) and toxic fluorine-containing gases (e.g., PF_5_, POF_3_, and HF) [25,26]. However, HF emission from PVDF was drastically suppressed by the addition of calcium salts [27,28]. It is, therefore, necessary to consider removing F from the recovered materials or to apply processes to these materials that tolerate the presence of F. 

Thus, we herein report the recovery of PGMs from spent autocatalysts based on the concept of “spent autocatalyst + waste LIB → resources”. In addition, we examine the immobilization of F on cordierite (2MgO·2Al_2_O_3_·5SiO_2_), which is a matrix component of autocatalysts, with the aim of realizing cleaner production.

## 2. Materials and Methods

Figure 1 shows the flowchart of the leaching process. Herein, PGM recovery experiments from spent autocatalysts were conducted using lithium carbonate (Li_2_CO_3_) and lithium fluoride (LiF) as Li sources.

### 2.1. Sample Preparation

#### 2.1.1. Starting Materials

A spent autocatalyst (NIST, Standard Reference Materials 2557 Used Auto Catalyst (Monolith)), Li_2_CO_3_ (FUJIFILM Wako Pure Chemical Corporation, Osaka, Japan, Wako Special Grade 122–01132), LiF (FUJIFILM Wako Pure Chemical Corporation, Osaka, Japan, Wako Special Grade 127–01785), and synthetic cordierite (2MgO·2Al_2_O_3_·5SiO_2,_ Marusu Glaze Co., Ltd., Aichi, Japan, SS-200, mean particle size of 7.5 µm) were used. The spent autocatalyst comprised monolith catalysts collected from vehicles in the 1990s, which were then crushed and sieved through a 200 mesh (<75 µm). Table 1 lists the concentrations of the various elements detected in the spent autocatalyst sample [29].

#### 2.1.2. Sample Preparation

Table 2 shows the conditions for sample preparation. Experimental samples were prepared by adding Li salts to synthetic cordierite or to the spent autocatalyst. Li_2_CO_3_ and LiF were employed as the Li salts.

After weighing the desired quantities of the aforementioned powders, the mixture was ground using an agate mortar and placed on an alumina boat (Sansyo Co., Ltd., Tokyo, Japan, SAB-995 AB-7). To prevent the Li salts from reacting with the alumina boat, a piece of gold foil was placed onto the boat, and the sample was then placed on top of it.

### 2.2. Calcination Experiments

Figure 2 shows a schematic of the calcination apparatus employed in this study. A mixture of the starting materials was inserted into a quartz tube installed in an electric furnace (Koyo Thermo Systems Co., Ltd., Nara, Japan, KTF030N1). Both ends of the quartz tube were plugged with silicon caps, and air was passed from one end to the other at a rate of 100 mL/min. The temperature profile consisted of two steps; in the first step, the temperature was raised to 100 °C at a rate of 10 °C/min and held at this temperature for 10 min. In the second step, the temperature was raised to various other temperatures between 600 and 900 °C. The sample was maintained at each temperature for 0.5–3 h. Table 2 summarizes the calcination conditions. After calcination, the interior of the furnace was cooled to <100 °C with a flow of air to obtain the calcined sample.

### 2.3. Leaching Experiments

#### 2.3.1. Hydrochloric Acid

The calcined sample was hand-crushed using an agate mortar, divided into two equal parts, and placed into two polytetrafluoroethylene (PTFE) inner-cylinder containers. A 12 M (37%) HCl solution (FUJIFILM Wako Pure Chemical Corporation, Osaka, Japan, Wako Grade 1 087–01071) was used to prepare 1.5 M, 3 M, and 6 M HCl solutions by diluting with ultrapure water. To each container, a HCl solution of the desired concentration (10 mL) was added. These containers were then covered and placed in a pressure-resistant stainless-steel outer cylinder, which was then placed in an oven (ADVANTEC, DRN320DD). The oven temperature was raised to 40 °C at a rate of 10 °C/min, maintained for 5 min, then further increased to 180 °C, and maintained at this temperature for 2 h. Subsequently, the inside of the oven was cooled to <100 °C under a flow of air prior to the removal of the reaction container. To suppress the reaction, the reaction container was rapidly cooled in a vat containing ice-water. The inside of the PTFE container was then washed with ultrapure water, and the solids were separated by filtering the reaction solution through a 0.2 µm PTFE membrane filter under reduced pressure. The filtrate volume was made up to 100 mL with a 2.5 % HCl solution to provide the leachate. The solid residues were then leached using aqua regia (see Section 2.3.2).

#### 2.3.2. Aqua Regia

A sample of the spent autocatalyst (1.0 g) and aqua regia (30 mL) were added to three tall beakers, and the beakers were heated on a hot plate at 150 °C for 1.5 h. After cooling, the solids were separated by filtering the solutions through a PTFE membrane filter (pore size = 0.2 µm) under reduced pressure. The leachate was made by diluting the filtrate volume to 100 mL using a 2.5% HCl solution. The solid residue was put into the beaker again to leach out the metals under the same leaching procedure. The leachate obtained after repeating this process five times was used for analysis.

### 2.4. Characterization

#### 2.4.1. Thermogravimetric–Differential Thermal Analysis

Thermogravimetric–differential thermal analysis (TG–DTA) measurements of the samples were performed using a TG/DTA6300 instrument (Hitachi High-Tech Science Corporation, Tokyo, Japan). The desired sample (10 mg) was charged to a ⌀5 mm alumina cell and heated to 1000 °C at a rate of 10 °C/min under a 300 mL/min flow of air. As a reference sample, *α*-Al_2_O_3_ powder was used.

#### 2.4.2. Elemental Analysis

The concentrations of the metals (Pt, Rh, Pd, and Li) in the leachate were quantified using inductively coupled plasma mass spectrometry (ICP-MS, ELEMENT 2, Thermo Fisher Scientific, Waltham, MA, USA) or inductively coupled plasma atomic emission spectroscopy (ICP-AES, SPECTRO ARCOS, SPECTRO Analytical Instruments, Kleve, Germany). The amount of each leached metal was calculated using Equation (1). The quantities of the PGMs in the solid sample were calculated on the basis of the concentrations of the elements detected in the spent autocatalyst (see Table 1).
(1)$$\mathrm{Leached}\;\mathrm{metal}\;\mathrm{amount}\;\left[\%\right]=\frac{\mathrm{Quantity}\;\mathrm{of}\;\mathrm{metal}\;\mathrm{in}\;\mathrm{leachate}}{\mathrm{Quantity}\;\mathrm{of}\;\mathrm{metal}\;\mathrm{in}\;\mathrm{solid}\;\mathrm{sample}}\times 100.$$

#### 2.4.3. XRD Analysis

The crystalline phase of the sample was identified by powder X-ray diffraction (XRD) analysis (Bruker Co., Ltd., Karlsruhe, Germany, D8 ADVANCE/L, USA). Measurements were taken at a voltage of 40 kV, a current of 40 mA, and 2*θ* values of 10–80° with a step width of 0.02° and a counting time of 1 s/step.

## 3. Results and Discussion

### 3.1. Calcination and Influence of Li_2_CO_3_ Addition

Figure 3 shows the TG–DTA curves obtained for the spent autocatalyst alone and a mixture of the spent autocatalyst and Li_2_CO_3_. In Figure 3a, which shows the results obtained using the spent autocatalyst alone, an increase in weight was observed over two temperature regions: the initial temperature to 180 °C and 560 °C to 770 °C. Although the weight decreased from 180 °C to 560 °C and above 770 °C, in all situations, the maximum change in weight was approximately ±0.2%. A very subtle peak attributed to an endothermic reaction close to 777 °C was also observed. In contrast, the weight of the sample containing Li_2_CO_3_ (Figure 3b) decreased by 5% between 545 °C and 735 °C; the endothermic peak observed at 720 °C was attributed to the melting of Li_2_CO_3_ (m.p. Li_2_CO_3_ = 723 °C) [30]. The exothermal reaction due to the formation of complex oxides could not be observed because the small amount of PGMs precluded detection.

Figure 4a,b show the XRD profiles obtained for the spent autocatalyst before calcination (condition I-A) and after calcination at 800 °C for 3 h (condition I-B). In both profiles, diffraction peaks corresponding to cordierite were observed. Furthermore, the enlarged image shown in Figure 4c indicates that the diffraction between 10° and 11° shifted toward a higher angle following calcination. As this peak originated from the 100 diffraction of cordierite, this observation indicates that calcination reduced the *a*-axis lattice constant. In addition, as shown in Figure 4c–e, new peaks were detected between 26.5° and 27°, and between 34° and 34.5° after calcination. Despite their low peak intensities, these findings suggest the generation of trace amounts of new compounds.

Figure 5 shows the results of PGM leaching by calcination and Li_2_CO_3_ addition. When an uncalcined sample (condition I-A) was leached using HCl alone, the amount of Pt leached was 4.5%, which was significantly lower than that achieved using aqua regia (91.7%). In contrast, the amount of Pt leached from a calcined sample at 800 °C for 3 h (condition I-B) increased to 84.3%. Moreover, the combination of Li_2_CO_3_ addition and calcination (condition I-C-1) resulted in the complete leaching of Pt.

Similarly, the amounts of Rh and Pd leached into HCl under condition I-A were 28.3% and 42.0%, respectively, which are less than half the quantities leached upon treatment with aqua regia. In contrast, the amounts of leached Rh and Pd under condition I-B increased to 76.9% and 75.9%, respectively, and they increased further to 86.5% and 85.0%, respectively, under condition I-C-1. These results revealed that the calcination with Li_2_CO_3_ effectively improved the Pt, Rh, and Pd leaching rates.

The increased amount of leached PGMs observed upon Li-salt addition was attributed to the generation of composite oxides that are soluble in HCl. Moreover, the majority of PGMs were found to be leached using HCl alone when sample calcination was carried out in advance, even in the absence of Li salts. Although the reason for this remains unclear, it is assumed to be due to the impact of compounds that can oxidize PGMs, such as Ce-oxide co-catalysts present in the spent autocatalysts.

### 3.2. Impact of Li_2_CO_3_ Concentration in Li Salt Mixture (Conditions I-C and II)

Figure 6 shows the amount of PGMs leached upon varying the ratio between Li_2_CO_3_ and LiF additives (conditions I-C-1, I-C-2, I-C-3, I-C-4, and I-C-5). As indicated, complete Pt leaching was achieved under condition I-C-1 (Li_2_CO_3_ concentration: 100 mass%). Although the amount of leached Pt slightly decreased as the amount of added LiF increased, the former remained sufficiently high (95.3%) even at a LiF concentration of 100 mass% (condition I-C-5). Because these values were all higher than those achieved when leaching was carried out using aqua regia, the effects of Li-salt addition were observed. In contrast, the maximum amounts of leached Rh and Pd were 86.5% and 86.1%, respectively, and there was no clear correlation between these amounts and the concentration of Li_2_CO_3_ in the Li-salt mixture. As seen in Figure 5, the leaching rates of Pt and Rh increased when using Li salts and HCl, while that of Pd remained unchanged. Moreover, a higher Pd leaching rate was achieved in aqua regia. 

The amount of leached Li was the highest under condition I-C-1 (97.9%) and tended to decrease as the ratio of Li_2_CO_3_/(Li_2_CO_3_ + LiF) decreased. Indeed, under condition I-C-5 (100 mass% LiF), this amount was reduced to 50.7%. We could not detect Li even after the aqua regia treatment of the solid residues remaining after HCl leaching. On the basis of these observations, we envisaged that approximately half of the added Li was converted to a form that is insoluble even in aqua regia.

Figure 7 shows the XRD profiles of calcined samples containing cordierite (conditions II-A, II-B, and II-C), while Figure 8 shows those of the residues after HCl leaching. After calcination of cordierite alone (Condition II-A, Figure 7a), the crystal structure was maintained, while MgAl_2_O_4_ was the main phase after leaching (Figure 8a). This showed that Si was separated from cordierite and that its crystal phase changed. When Li_2_CO_3_ was added (condition II-B, Figure 7b), Li_2_O(Al_2_O_3_)SiO_2_, Li_4_SiO_4_, and MgAl_2_O_4_ were generated after calcination. The diffraction peaks of these Li complex oxides disappeared after leaching, and only MgAl_2_O_4_ was detected in the crystalline phase (Figure 8b). Although they were not detectable through XRD analysis because of their low concentrations, Pt, Rh, and Pd produced complex oxides with Li that are soluble in HCl, which in turn improved the leaching rates of the PGMs. Regardless of Li salt addition, the residues showed a wide range of diffraction peaks from 17° to 27°. This suggests that amorphous components are present in the sample. According to the positions of the diffraction peaks, these may originate from an amorphous Si compound [31].

Even in the case of the sample where LiF was added (condition II-C, Figure 7c), cordierite decomposed during calcination to give LiAlSiO_4_ and MgF_2_. Although the rate of Li leaching by HCl decreased with the addition of LiF (Figure 6), this may have been due to the incorporation of Li into a compound that is sparingly soluble in HCl, e.g., MgF_2_. In contrast, the PGM leaching rate did not significantly decrease even when the type of Li salt was changed. On the basis of the observations, it was inferred that the formation of PGM-containing complex oxides proceeded regardless of the type of Li salt employed.

### 3.3. Effects of Various Parameters on PGM Leaching

#### 3.3.1. HCl Concentration and the Type of Li Salt (Conditions I-C-1 and I-C-5)

Figure 9 shows the relationship between the HCl concentration and the leaching rates of various metals. When the HCl concentration was increased to 1.5, 3, 6, and 12 M, the leaching rates of Pt from the samples with added Li_2_CO_3_ (condition I-C-1) increased to 62.9%, 84.6%, 87.4%, and 104.2%, respectively. For the same increase in HCl concentration, the leaching rates of Pt from the samples with added LiF (condition I-C-5) increased to 78.7%, 83.1%, 83.0%, and 95.3%, respectively. Although the leaching rate of Pt increased with increasing HCl concentration, there was no significant difference in the leaching rate at HCl concentrations of 3 and 6 M. Furthermore, except at 1.5 M HCl, the leaching rates increased with increasing Li_2_CO_3_/(Li_2_CO_3_ + LiF) ratio. The leaching behavior of Pd was similar to that of Pt. The leaching rates of Rh increased remarkably with HCl concentration irrespective of the type of Li salt.

Through the calcination of the Li salt and the use of spent autocatalysts, the leaching rates of PGMs into HCl drastically increased compared to those of the untreated samples [12]. This is due to the generation of Li–PGM complex oxides that are soluble in HCl. Under condition I-C-1, Pt, Pd, and Rh are considered to convert to Li_2_PtO_3_, Li_2_PdO_2_, and LiRhO_2_, respectively. As previously described, the leaching rate of Pt from Li_2_PtO_3_ increased with HCl concentration [9]. This is consistent with the results obtained in this study. Although Li_2_PdO_2_ easily dissolves in low-concentration HCl at ambient temperatures [11], under the conditions examined in our study, the complete leaching of Pd was not possible. This may have been due to the incorporation of Pd in the host materials that is less soluble than Li_2_PdO_2_. Although the leaching rate of Rh significantly increased with increasing HCl concentration, it did not completely leach, similar to the case of Pd. This was because some Rh remained unreacted. Although the calcination of Rh powder and Li_2_CO_3_ generates LiRhO_2_, some unreacted Rh remains after calcination at 800 °C for 3 h [13]. Since PGMs rarely dissolve in HCl in the metallic state, the extent of dissolution decreases if unreacted PGMs exist.

The leaching rates of Li varied between 96.1% and 98.3% for the samples using Li_2_CO_3_ and between 46.7% and 50.7% for those using LiF alone, and they were almost constant regardless of the HCl concentration. Because the calcination products varied depending on the Li salt added, the type of decomposition products is believed to reflect the leaching behavior of Li. In other words, Li was almost completely leached from the products (i.e., from Li_2_O(Al_2_O_3_)SiO_2_ and Li_4_SiO_4_) when Li_2_CO_3_ was used. In contrast, when LiF was added, MgF_2_ remained even after HCl leaching; it was likely that some Li was also present.

#### 3.3.2. Calcination Temperature (Conditions I-D-1, I-D-2, I-D-3, and I-D-4)

Figure 10 shows the leaching rates of the metals from samples obtained by changing the calcination temperature (conditions I-D-1, I-D-2, I-D-3, and I-D-4). Almost complete leaching of PGM and Li was possible upon calcination between 600 and 900 °C. More specifically, the leaching rates were 89.6–100% for Pt, 79.1–86.5% for Rh, 81.7–85.0% for Pd, and 97.9% for Li. For both Li and Pd, the leaching rates were mostly maintained at these calcination temperatures. In contrast, a maximum difference of 10% was observed in the leaching rates of Pt. All PGMs showed maximum leaching rates from the samples calcined at 800 °C; the entire amount of Pt, 85.0% of Rh, and 86.5% of Pd were leached under this condition.

#### 3.3.3. Calcination Time (Conditions I-E-1, I-E-2, I-E-3, and I-E-4)

Figure 11 shows the leaching rates of the metals from the samples obtained by varying the calcination time at 800 °C (conditions I-E-1, I-E-2, I-E-3, and I-E-4). Pt and Li were completed leached after 1, 2, and 3 h of calcination, while 79.2% and 82.1% of Rh was leached after 0.5 and 1 h, respectively. However, the leaching rate of Rh reached its maximum, i.e., 94.9%, after 2 h. In addition, the amount of Pd leached after 0.5 h was 83.7%, and this increased to 94.7% and 97.5% after 1 and 2 h, respectively. The reduced leaching rate after 3 h calcination could be attributed to enhanced sintering and melting. Under the conditions employed in this study, 2 h of calcination at 800 °C was concluded to be the optimum condition for the leaching of PGMs.

## 4. Conclusions

In this study, the recovery of PGMs from spent automotive catalysts (autocatalysts) was attempted using Li salts and HCl according to the concept of “spent autocatalyst + waste LIBs → resources”. As a result, the complete leaching of Pt from samples containing added Li_2_CO_3_ was achieved using HCl. Under the same conditions, the amounts of Rh and Pd leached were 94.9% and 97.5%, respectively. Upon the addition of LiF to the samples, the amount of leached Pt decreased by approximately 5%, but there was no significant change in the amounts of leached Rh and Pd. XRD analysis revealed that a complex oxide—Li_4_SiO_4_—was generated when Li_2_CO_3_ was added to the samples, and that LiAlSiO_4_ formed when LiF was added. Therefore, a close relationship was confirmed between the generation of these compounds and the leaching behavior of Li. Furthermore, the immobilization of F on cordierite, which is a matrix component of the autocatalysts, was observed. Mg present in cordierite reacted with F^−^ ions in solution to generate MgF_2_, which plays a significant role in the immobilization of F. Optimal leaching efficiencies were obtained through calcination at 800 °C for 2 h. The leaching efficiency increased with HCl concentration and reached the maximum at 12 M HCl.

## Figures and Tables

**Figure 1 materials-14-06843-f001:**
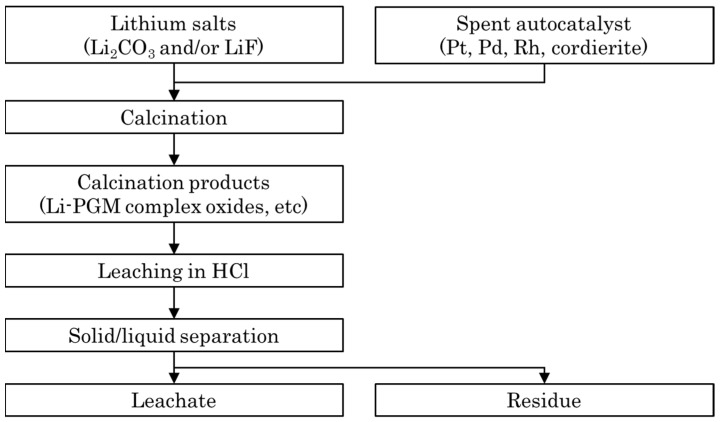
Flowchart of the platinum group metal (PGM) leaching process using Li salts.

**Figure 2 materials-14-06843-f002:**
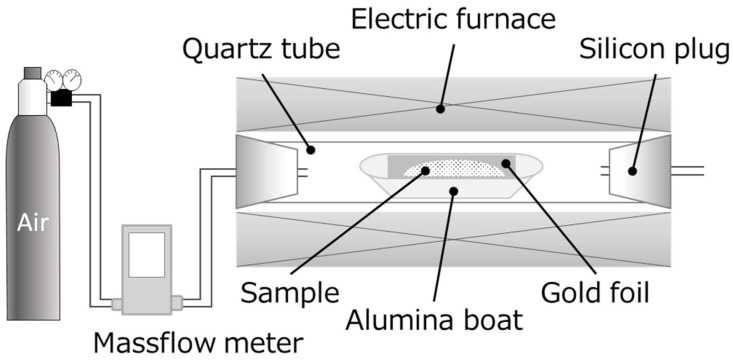
Schematic of the calcination apparatus.

**Figure 3 materials-14-06843-f003:**
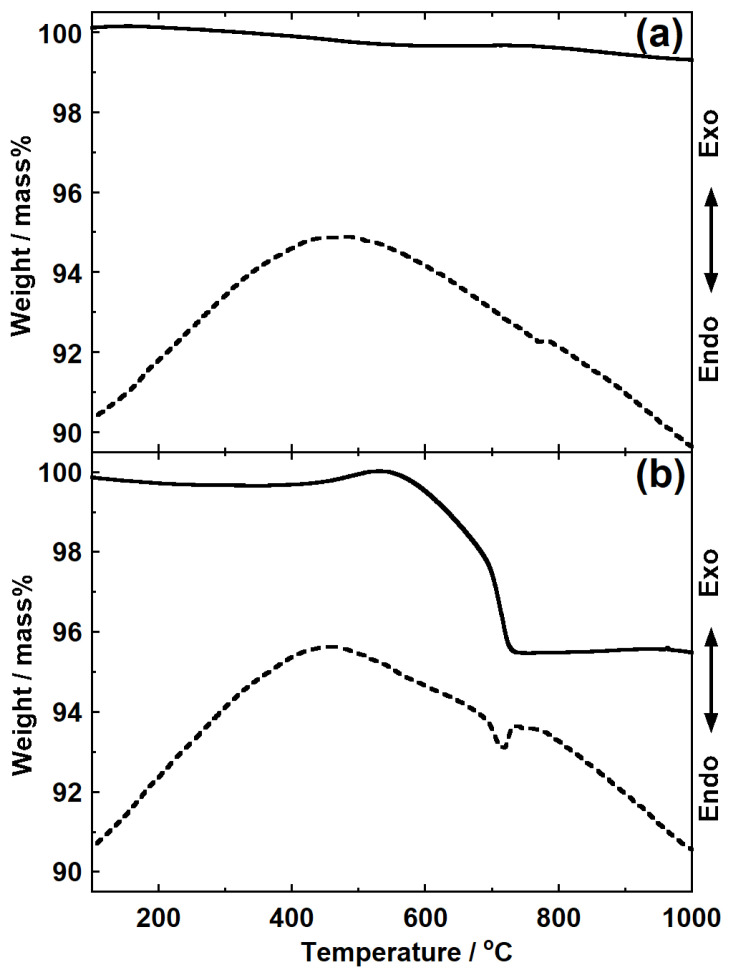
Thermogravimetric–differential thermal analysis (TG–DTA) profiles of samples obtained using (**a**) the spent autocatalyst alone and (**b**) the spent autocatalyst with Li_2_CO_3_. Solid lines: TG curves; dashed lines: DTA curves.

**Figure 4 materials-14-06843-f004:**
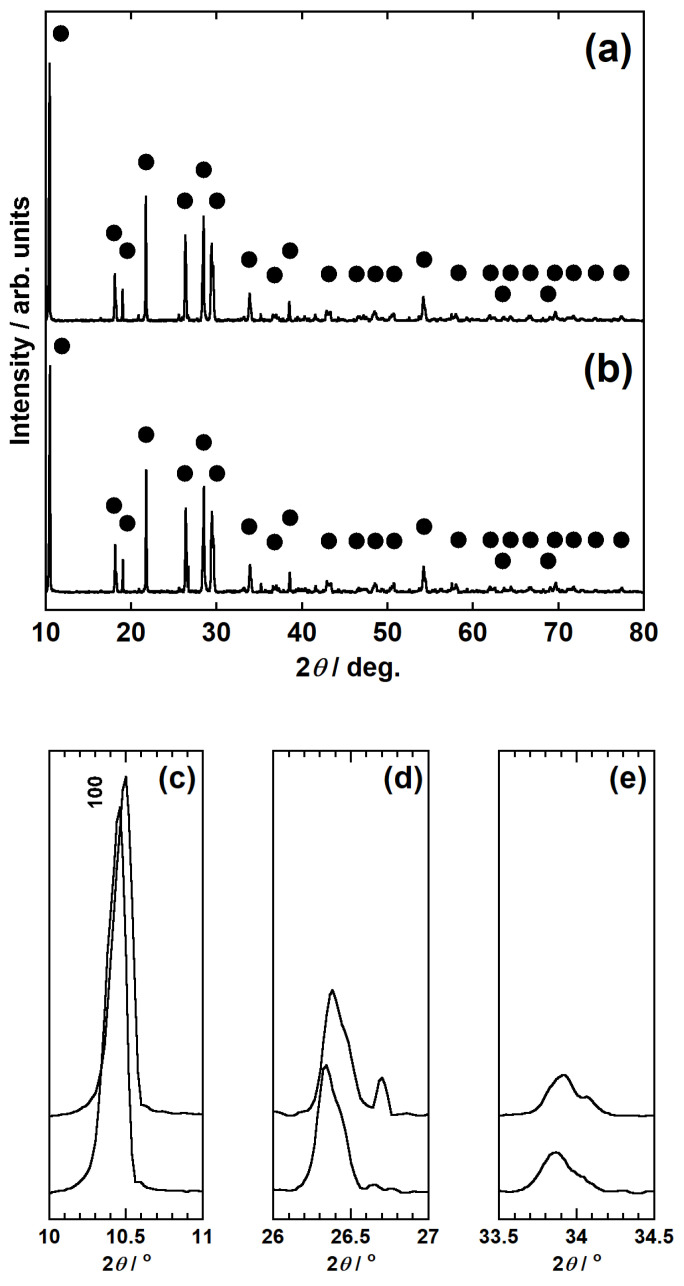
X-ray diffraction (XRD) profiles of the spent autocatalyst samples obtained (**a**) before calcination (condition I-A) and (**b**) after calcination at 800 °C for 3 h (condition I-B). The black circles indicate the cordierite phase. (**c**–**e**) Enlarged XRD profiles of the spent autocatalyst samples. Lower plot: condition I-A; upper plot: condition I-B.

**Figure 5 materials-14-06843-f005:**
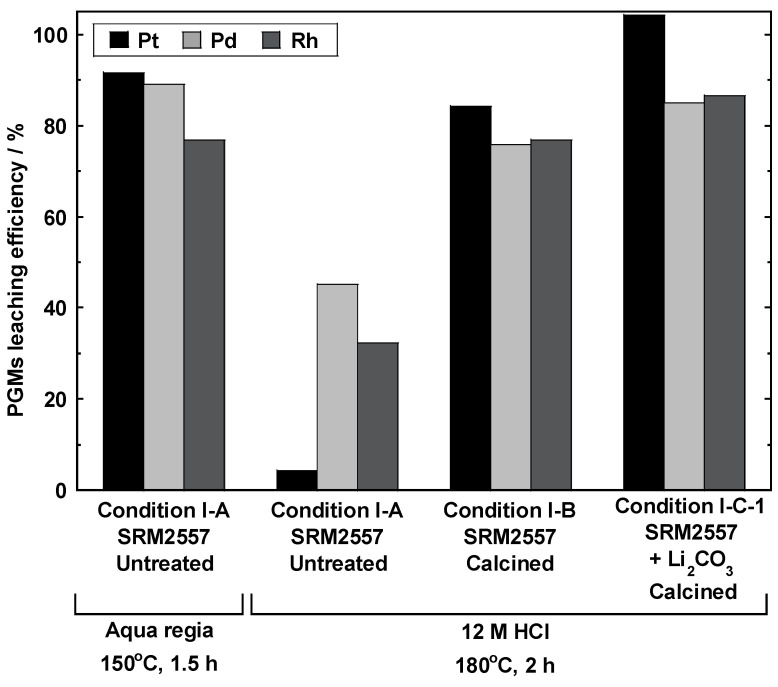
Pt, Pd, and Rh leaching efficiencies in the solutions obtained under conditions I-A, I-B, and I-C-1.

**Figure 6 materials-14-06843-f006:**
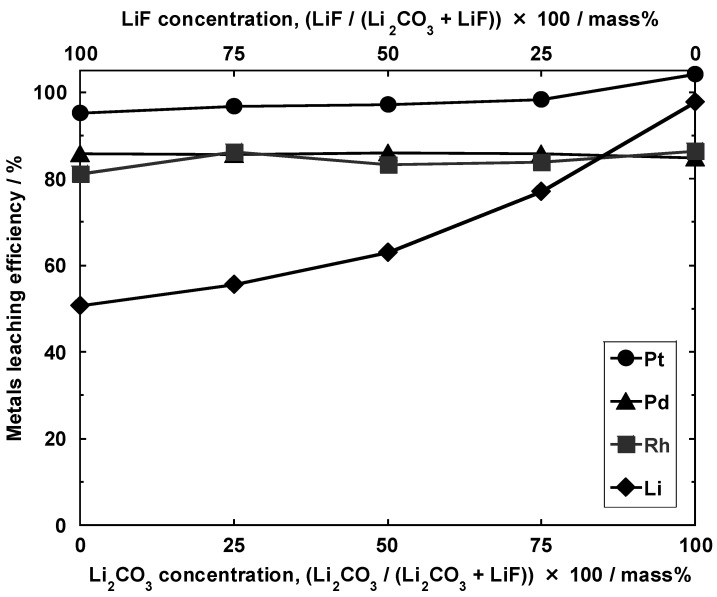
Pt, Pd, Rh, and Li leaching efficiencies in the solution samples obtained under conditions I-C-1, I-C-2, I-C-3, I-C-4, and I-C-5. Dissolution conditions: 12 M HCl at 180 °C for 2 h.

**Figure 7 materials-14-06843-f007:**
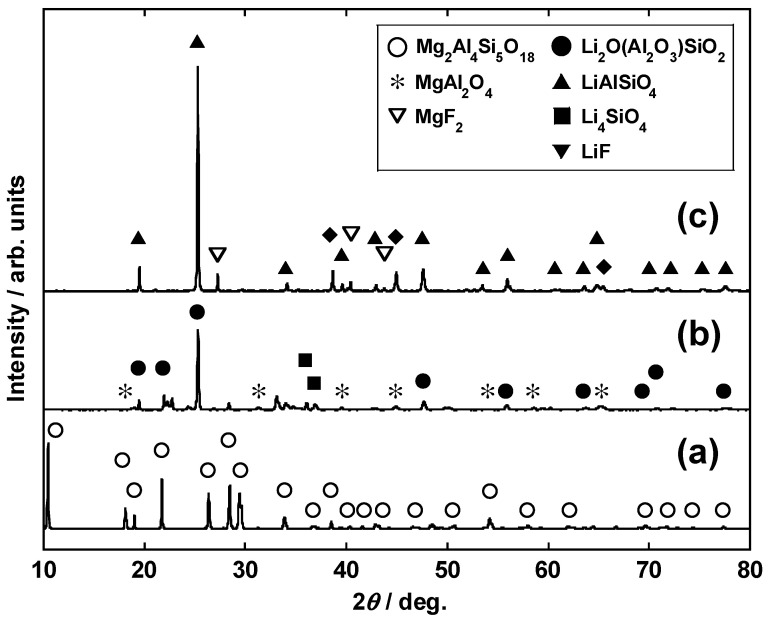
XRD profiles of the cordierite-containing calcined samples obtained under conditions (**a**) II-A, (**b**) II-B, and (**c**) II-C.

**Figure 8 materials-14-06843-f008:**
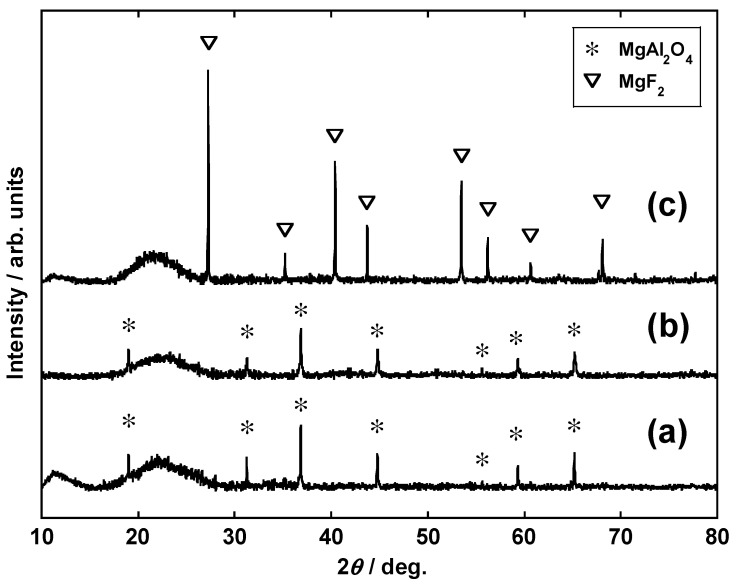
XRD profiles of the residues obtained under conditions (**a**) II-A, (**b**) II-B, and (**c**) II-C. Dissolution conditions: 12 M HCl at 180 °C for 2 h.

**Figure 9 materials-14-06843-f009:**
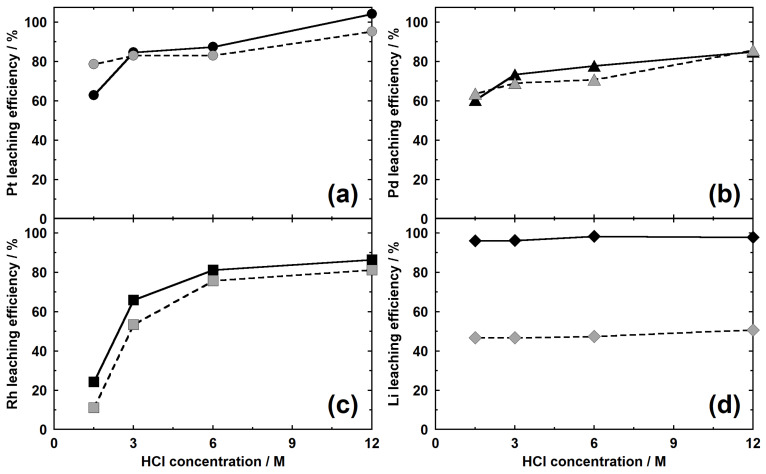
Metal leaching efficiencies of the samples obtained under conditions I-C-1 and I-C-5. (**a**) Pt, (**b**) Pd, (**c**) Rh, and (**d**) Li. Solid lines: condition I-C-1, dashed lines: condition I-C-5.

**Figure 10 materials-14-06843-f010:**
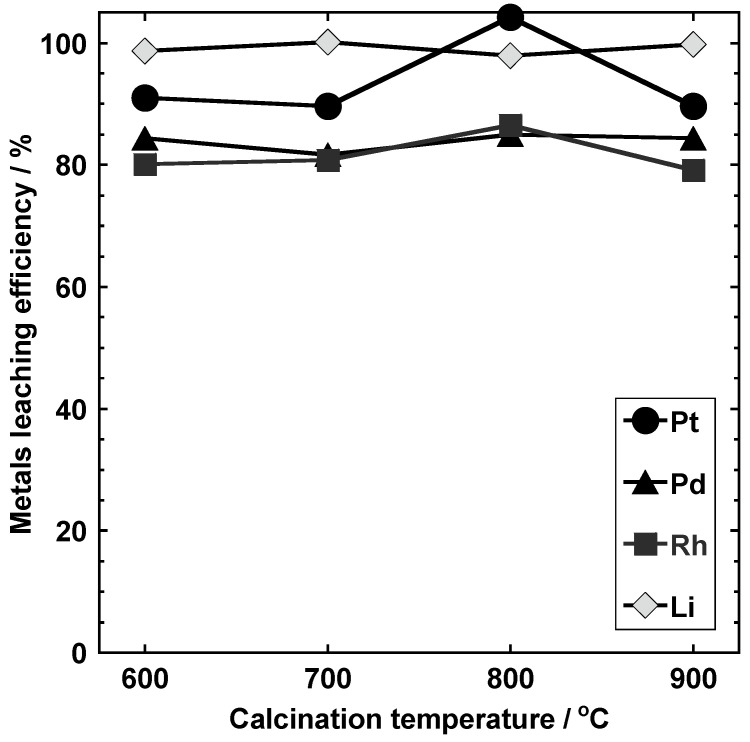
Pt, Pd, Rh, and Li leaching efficiencies of the samples obtained under conditions I-D-1, I-D-2, I-D-3, and I-D-4.

**Figure 11 materials-14-06843-f011:**
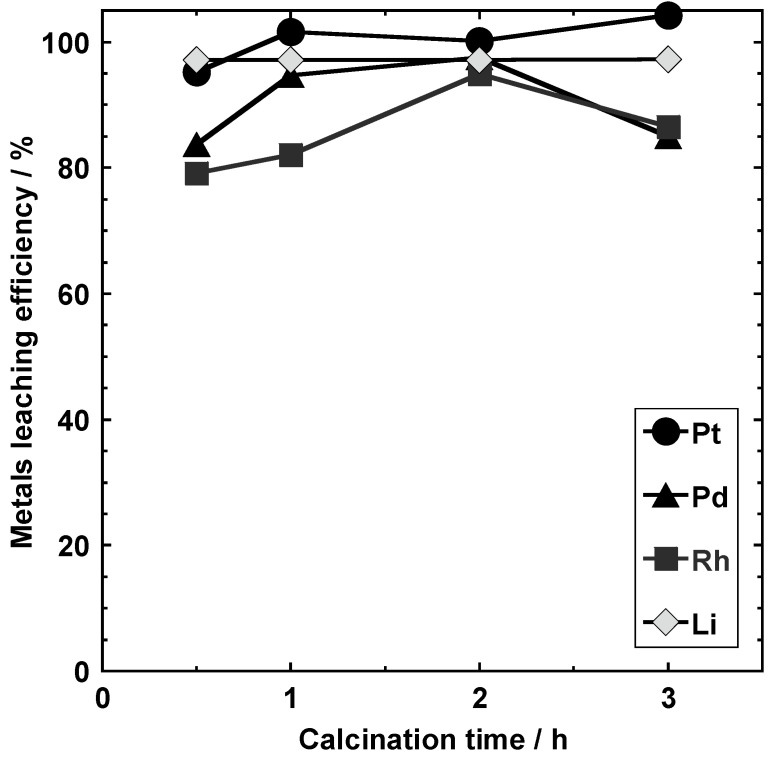
Pt, Pd, Rh, and Li leaching efficiencies of the samples obtained under conditions I-E-1, I-E-2, I-E-3, and I-E-4.

**Table 1 materials-14-06843-t001:** Concentrations of elements present in the spent autocatalyst sample [29].

Element	Concentration (mg/kg)	Element	Concentration (mass%)
Pt	1131 ± 11	Al	20
Pd	233.2 ± 1.9	Si	18
Rh	135.1 ± 1.9	Mg	6
Pb	13,931 ± 97	Fe	1.5
		Ce	1.3
		Ni	0.5
		Ba	0.29
		Ca	0.2
		La	0.07

**Table 2 materials-14-06843-t002:** Quantities of reagents added and subsequent calcination conditions.

Preparation Conditions	Starting Materials				Calcination Temperature (°C)	Holding Time (h)
	Spent Autocatalyst (g)	Cordierite (g)	Li_2_CO_3_ (mg)	LiF (mg)		
I-A	1.0	-	-	-	-	-
I-B	1.0	-	-	-	800	3
I-C-1 ^1^	1.0	-	100	-	800	3
I-C-2	1.0	-	75	25	800	3
I-C-3	1.0	-	50	50	800	3
I-C-4	1.0	-	25	75	800	3
I-C-5	1.0	-	-	100	800	3
I-D-1	1.0	-	100	-	600	3
I-D-2	1.0	-	100	-	700	3
I-D-3 ^1^	1.0	-	100	-	800	3
I-D-4	1.0	-	100	-	900	3
I-E-1	1.0	-	100	-	800	0.5
I-E-2	1.0	-	100	-	800	1
I-E-3	1.0	-	100	-	800	2
I-E-4 ^1^	1.0	-	100	-	800	3
II-A	-	1.0	-	-	800	3
II-B	-	1.0	500	-	800	3
II-C	-	1.0	-	500	800	3

^1^ These sets of conditions were the same and gave the same sample.

## Data Availability

The data can be provided by the corresponding author on request.

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
