# Peer review of "Recovery of Platinum Group Metals from Spent Automotive Catalysts Using Lithium Salts and Hydrochloric Acid"

_materials, 2021, doi:10.3390/ma14226843_

Round 1

Reviewer 1 Report

The authors use the "spent autocatalyst + waste LIBs → resources" technique to recovery of platinum group metals. In my opinion, the manuscript is very interesting, showing results clearly explained and potentially useful in the research field.

Few comments or considerations for the authors from me:

Could the authors explain gains/losses of the mass in the thermogravimetric–differential thermal analysis (TG–DTA) profiles? It is not clear the differences between the spent autocatalyst alone and the spent autocatalyst with Li2CO3. What is the information that the authors know? What information do the authors want to know? Please, the aim purpose of these tests.

Figure 4 should be improved. Inset c to e show enlarged XRD patterns, but the information from XRD technique cannot be deduced if the results are displayed like this.

In the conclusions section, a line explaining the optimal conditions (quantities, temperature ...) would be grateful.

After these small considerations, the manuscript could be considered for publication in Materials Journal.

Reviewer 2 Report

PGM recovery from secondary sources is important topic for future economy. It is known, that nowadays large portion of PGMs can be recovered. New effective protocols are always iportant for further progress in the field. Authors address important topic of recovery of PGMs from autocatalysts. Topic is potentially interesting for broader audience. Nevertheless, I guess, topic is out of scope of mdpi Materials and would be mmore suitable for other more "chemical" journals. Beside this, there are several issues which should be clarified before considering for publication.

1. In which chemical form PGMs are present in the original materials: metallic nanopartiucles, oxides, chlorides? I would suggest to perform fist the study with model systems or even with mmetallic powders to show that technique can be applied for other systems.

2. Authors discuss calcination with Li basic salts. Oxidation of PGMs by air / oxygen in the presence of NaOH or similar basic agents is well-known industrial process. Nevertheless, authors do not discuss any role of air in the current process. I would suggest to give more explanationns of the role of oxidasing agent in the current process.

3. After leaching, in which form PGMs are present? PGMs solutions after leaching seem to be quite diluted. I would expect a discussion of further processinng of such solutions to understand if it is practically possible (if possible, how) to extract PGMs from resulting solutions after leaching.

4. Pd is usually more chemically active, inn comparison with Pt. Why extraction of Pt is higher in compariwon with Pd?

Page 7. I suggest to give a TG profile for pure Li2CO3 for correct comparison.

Line 216: "from the 100 diffraction" thee meaning is not really clear

Fig. 4: I suggest to give complete phase analysis and show marks for each phase on the plot.

Fig. 5: ammount of Pt extracted from the Li2CO3-calcinated mixtures seem to be above 100 %. (similar for Fig. 6, Fig. 9a, Fig 10 at 800°C and Fig 11)

Round 2

Reviewer 2 Report

Authors addressed all issues mentioned in review.